# ReToMe-VA: Recursive Token Merging for Video Diffusion-based Unrestricted Adversarial Attack

## ABSTRACT

Recent diffusion-based unrestricted attacks generate imperceptible adversarial examples with high transferability compared to previous unrestricted attacks and restricted attacks. However, existing works on diffusion-based unrestricted attacks are mostly focused on images yet are seldom explored in videos. In this paper, we propose the Recursive Token Merging for Video Diffusion-based Unrestricted Adversarial Attack (ReToMe-VA), which is the first framework to generate imperceptible adversarial video clips with higher transferability. Specifically, to achieve spatial imperceptibility, ReToMe-VA adopts a Timestep-wise Adversarial Latent Optimization (TALO) strategy that optimizes perturbations in diffusion models' latent space at each denoising step. TALO offers iterative and accurate updates to generate more powerful adversarial frames. TALO can further reduce memory consumption in gradient computation. Moreover, to achieve temporal imperceptibility, ReToMe-VA introduces a Recursive Token Merging (ReToMe) mechanism by matching and merging tokens across video frames in the self-attention module, resulting in temporally consistent adversarial videos. ReToMe concurrently facilitates inter-frame interactions into the attack process, inducing more diverse and robust gradients, thus leading to better adversarial transferability. Extensive experiments demonstrate the efficacy of ReToMe-VA, particularly in surpassing state-of-the-art attacks in adversarial transferability by more than 14.16% on average.

## CCS CONCEPTS

• **Computing methodologies** → **Computer vision**.

## KEYWORDS

action recognition, unrestricted adversarial attacks, diffusion models

## 1 INTRODUCTION

Recent years have witnessed remarkable performance exhibited by Deep Neural Networks (DNNs) across various computer vision and multimedia tasks [8, 13]. However, the emergence of adversarial examples has posed a challenge to the robustness of DNNs [11]. These adversarial examples, created by making imperceptible modifications to benign samples, can easily deceive state-of-the-art DNNs. Importantly, adversarial examples generated against one

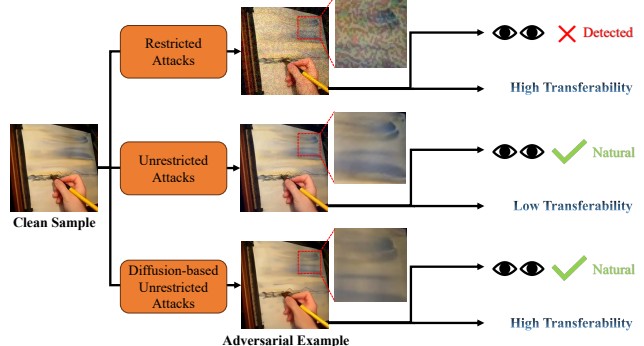

**Figure 1: Difference between restricted attacks, unrestricted attacks, and diffusion-based unrestricted attacks.**

model can also mislead other models even with different architectures [6, 36]. The transferability of adversarial examples makes it feasible to carry out black-box attacks, which highlight security flaws in safety-critical scenarios, such as face verification [30] and surveillance video analysis [6], etc. To avoid potential risks, it is crucial to expose as many "blind spots" of DNNs by deeply exploring the transferability of adversarial examples.

Nowadays, the majority of transfer-based adversarial attacks [21, 38, 39] try to guarantee "subtle perturbation" by limiting the $L_p$-norm of the perturbation (a.k.a. restricted attacks). However, adversarial examples generated under $L_p$-norm constraint have human-perceptible perturbations, thereby rendering them more easily detectable [1, 46]. Therefore, unrestricted adversarial attacks [43, 45], which optimize unrestricted but natural changes (such as texture, style, color modifications, etc.) for given benign samples, are beginning to emerge. These unrestricted attacks yield more imperceptible perturbations but fall short in transferability compared to restricted attacks. With diffusion models drawing significant attention, recent works [5, 7] have employed diffusion models for unrestricted attacks to generate imperceptible adversarial examples with high transferability. The difference between previous unrestricted attacks, restricted attacks, and diffusion-based unrestricted attacks is displayed in Figure 1. Nevertheless, existing works on diffusion-based unrestricted attacks are mostly focused on images yet are seldom explored in videos.

This paper investigates transferable diffusion-based unrestricted attacks across different video recognition models. Specifically, we map each frame into the latent space and optimize the latents along the adversarial direction. The challenge of video diffusion-based unrestricted attacks comes from three aspects. Firstly, given the fact that diffusion models tend to add coarse semantic information in the early denoising steps [22], premature manipulation of the latents from previous work [7] yields significant alternations to the crafted frames compared to the corresponding benign frames.

Concurrently, these spatial perceptible changes further result in temporal inconsistency in crafted adversarial videos when directly applying such generation to each frame. Consequently, further effort is needed to generate adversarial videos with temporal imperceptibility. Secondly, separately perturbing each benign frame induces monotonous gradients because the interactions among the video frames have not been fully exploited. Therefore, inter-frame interaction is necessary for boosting adversarial transferability. Lastly, the previous generation involves the gradient calculation throughout the entire denoising process, leading to a heavy memory overhead, especially when updating all the frames simultaneously.

To this end, we propose ReToMe-VA, which is the first video diffusion-based unrestricted adversarial attack framework, aiming at producing imperceptible adversarial video clips with higher transferability, as shown in Figure 2. Specifically, to achieve spatial imperceptibility, we introduce a Timestep-wise Adversarial Latent Optimization (TALO) that gradually updates perturbations in the latent space at each denoising timestep. Instead of calculating gradients of the entire denoising process, TALO only involves one timestep gradient calculation thereby reducing memory consumption in gradient computation. Furthermore, to reduce the spatial structure differences between benign and adversarial frames, TALO establishes constraints on the self-attention maps, which have been demonstrated to regulate structure effectively [5]. To effectively trade-off between spatial imperceptibility and adversarial transferability, TALO introduces the incremental iteration strategy, which prioritizes fewer iterations during the early timesteps to preserve the structure and increases the number of iterations during later timesteps to add more adversarial content. Therefore, TALO offers iterative and accurate updates to generate more powerful adversarial frames. To achieve temporal imperceptibility of adversarial video, we propose a novel Recursive Token Merging (ReToMe) mechanism, which recursively aligns tokens across frames according to the correlation and compresses the temporally redundant tokens to facilitate joint self-attention. With shared tokens in the self-attention module, ReToMe fixes the misalignment of details in per-frame optimization, resulting in temporally consistent adversarial videos. Additionally, inter-frame interaction can make the gradient of the current frame fuse information from associated frames, which has the potential to generate robust and diverse update directions to fool various target video models [34]. The ReToMe facilitates inter-frame interactions into the attack process, thus boosting the adversarial transferability.

Our contributions can be summarized as follows:

- We introduce the first framework for video diffusion-based unrestricted adversarial attacks, leveraging the Stable Diffusion model to generate imperceptible adversarial video clips with higher transferability.
- We propose a Timestep-wise Adversarial Latent Optimization strategy to achieve spatial imperceptibility. Besides, our novel recursive token merging mechanism maximally merges self-attention tokens across frames, thereby boosting adversarial transferability while achieving temporal imperceptibility.
- We conduct extensive experiments on video recognition models trained on both CNNs and Vits, as well as various

defense methods. Our results demonstrate that ReToMe-VA surpasses the best baseline by an average of 14.16% and 17.32%, respectively.

## 2 RELATED WORK

As there are no previous works focusing on transferable video unrestricted attacks, this section reviews recent works on transferable unrestricted attacks against image models and transferable restricted attacks against video models.

### 2.1 Transferable Image Unrestricted Attacks

In the transferable image unrestricted attacks, color manipulation-based approaches play a significant role. Semantic Adversarial Examples (SAE) [15] converts the image from the RGB color space to the HSV color space, followed by random perturbation of both the H (Hue) and S (Saturation) channels. ReColorAdv [17] optimizes color transformation within the CIELUV color space, employing a flexibly parameterized function 'f' to recolor every pixel color 'c' to a new one. Colorization Attack (cAdv) [3] utilizes a pre-trained colorization network for color transformation, simultaneously adjusting input hints and masks to generate more natural adversarial examples. Unlike the previous one, Adversarial Color Enhancement (ACE) [45] generates adversarial images by using and optimizing a simple piece-wise linear differentiable color filter, with fewer parameters and better performance. To prevent human detection of unrestricted disturbances, ColorFool [29] manually selects four human-sensitive semantic classes and modifies colors within these sensitive regions constrainedly in the Lab color space. To make adversarial images more natural, Natural Color Fool (NCF) [43] constructs a "distribution of color distributions" for different semantic classes based on an existing dataset, using fused color distribution and optimizable transfer matrix to generate adversarial images.

Except for color manipulation-based methods, Texture Attack (tAdv) [3] fuses the texture of images from another class to generate adversarial examples, with an additional constraint on the victim image to prevent producing artistic images. Different from Texture Attack, Adversarial Content Attack (ACA) [7] introduces a diffusion model to perform unrestricted attacks on image models. By leveraging the diffusion model as a low-dimensional manifold, ACA maps the victim image into the latent space, where adversarial attacks and optimizations are conducted. When compared to both color manipulation-based methods and texture attacks, ACA demonstrates superior capability in generating natural adversarial image examples by harnessing the powerful generative capacity of diffusion models. Therefore, this paper investigates the potential of leveraging the diffusion model to perform transferable video unrestricted attacks.

### 2.2 Transferable Video Restricted Attacks

In the transferable video restricted attacks, Temporal Translation (TT) [37] is a representative method, which prevents overfitting the surrogate model by optimizing adversarial perturbations over a set of temporal translated video clips, to enhance the transferability of video adversarial examples across different video models. Most recently, based on the observation that the intermediate features between image models and video models are somewhat similar [38],

**Figure 2: Framework overview of the proposed ReToMe-VA. For a video clip, DDIM inversion is applied to map the benign frames into the latent space. Timestep-wise Adversarial Latent Optimization is employed during the DDIM sampling process to optimize the latents. Throughout the whole pipeline, Recursive Token Merging and Recursive Token Unmerging Modules are integrated into the diffusion model to enhance its effectiveness. Additionally, structure loss is utilized to maintain the structural consistency of video frames. Ultimately, the resulting adversarial video clip is capable of deceiving the target model.**

some transferable cross-modal attacks from images to videos have emerged. For instance, Image To Video (I2V) [38] generates adversarial video clips on the ImageNet pre-trained model by minimizing the cosine similarity between intermediate features of each benign frame and its adversarial frame. However, I2V treats a video clip as an orderless image set and ignores the inherent temporal information in video clips. In contrast, Global-Local Characteristic Excited Cross-Modal Attack [34] fully considers video characteristics from both global and local perspectives, which performs global inter-frame interactions in the attack process to induce more diverse and stronger gradients and proposes local correlation disturbance to prevent the target video model from capturing valid temporal clues. Furthermore, Generative Cross-Modal Attack (GCMA) [6] trains perturbation generators against the ImageNet domain but can fool target models from video domains, which proposes a random motion module and a temporal consistency loss based on intermediate features to narrow the gap between the image and video domains. Different from all of the prevision works that focus on restricted attacks, this work studies unrestricted attacks on video models.

## 3 METHODOLOGY

### 3.1 Diffusion-based Unrestricted Attack Framework

Given a benign video clip $x \in \mathcal{X} \subset \mathbb{R}^{N \times H \times W \times C}$ with $N$ frames $\{x^1, x^2, ..., x^N\}$ and its corresponding ground-truth label $y \in \mathcal{Y} = \{1, 2, ...K\}$, where $N, H, W, C$ denote the number of frames, height, width and the number of channels respectively, $K$ denotes the number of classes. Let $F_\theta$ denote the video recognition model trained on the video dataset $\mathcal{X}$. We use $F_\theta(x) : \mathcal{X} \rightarrow \mathcal{Y}$ to denote the prediction of the video recognition model $F_\theta(x)$ for $x$. Our goal is to craft unrestricted adversarial video clip $\hat{x}$ against a surrogate video recognition model $G\phi$ leveraging the Stable Diffusion [28] to deceive the target video recognition model $F_\theta$.

Prior works on image diffusion-based unrestricted attacks [5, 7] use the DDIM inversion [23] technology to map the benign image back into the diffusion latent space by reversing the deterministic

sampling process, then optimize the latent of the image along the adversarial direction. Finally, the adversarial image is generated from the optimized adversarial latent through the entire denoising process. For simplicity, the encoding and decoding of the VAE is ignored, as it is differentiable. However, such generation has obvious limitations for video attacks when applied directly to each frame. Firstly, given the fact that diffusion models tend to add coarse semantic information during the early denoising steps [22], premature manipulation tends to change the layouts or semantic structure of frames, which leads to semantic inconsistency and changes. This spatial inconsistency further leads to temporal inconsistency in adversarial videos. Furthermore, because this framework applied in video attacks involves updating all the frames simultaneously, the gradient calculation throughout the entire denoising process leads to a heavy memory overhead and large time consumption.

Therefore, we propose our ReToMe-VA to address these challenges, as shown in Figure 2. Specifically, we utilize the Timestep-wise Adversarial Latent Optimization (Sec.3.2) in the denoising process and introduce a Recursive Token Merging (Sec.3.3) technique to maintain the temporal consistency and boost adversarial transferability. The algorithm of ReToMe-VA is presented in Algorithm 1.

### 3.2 Timestep-wise Adversarial Latent Optimization

Existing latent optimization approaches which update latent at a fixed timestep are usually insufficiently flexible and stable in controlling the generation of adversarial video clips, therefore we propose Timestep-wise Adversarial Latent Optimization (TALO) to gradually update perturbations in the latent space at each denoising timestep. After the inversion of the DDIM, we obtain the reversed latents $\{x_0, x_1, ..., x_T\}$ from timestep 0 to $T$, where $x_0$ is $x$. For the trade-off between imperceptibility and adversarial transferability, we start adversarial optimization from the latent $x_{t_s}$ at $t_s$ timestep rather than from Gaussian noise at $T$ timestep. We denote $\hat{x}_t$ as the adversarial latents at $t$ timestep, we initialize $\hat{x}_{t_s} = x_{t_s}$. At each timestep $t$ of denoising, we predict the final output $\hat{x}_0^t$ for each

frame to substitute the adversarial output $\hat{x}_0$ for the prediction of the surrogate model $G_\phi$. The calculation of $\hat{x}_0^t$ and our adversarial objective function is expressed as follows:

$$\hat{x}_0^t = \frac{\hat{x}_t - \sqrt{1 - \alpha_t}\epsilon_\theta(\hat{x}_t, t)}{\sqrt{\alpha_t}} \quad (1)$$

$$\arg\min_{\hat{x}_t} \mathcal{L}_{attack} = -J(\hat{x}_0^t, y, G\phi) \quad (2)$$

where $\alpha_t$ represents the parameters of the scheduler, $\epsilon_\theta$ denotes the noise predicted by the UNet, and $J(\cdot)$ is the cross-entropy loss. After optimizing latents $\hat{x}_t$, we generate a sample $\hat{x}_{t-1}$ from $\hat{x}_t$ for the preparation of next timestep-wise optimization via:

$$\hat{x}_{t-1} = \sqrt{\alpha_{t-1}} \left( \frac{\hat{x}_t - \sqrt{1 - \alpha_t}\epsilon_\theta(\hat{x}_t, t)}{\sqrt{\alpha_t}} \right) \\ + \sqrt{1 - \alpha_{t-1} - \sigma_t^2}\epsilon_\theta(\hat{x}_t, t) \quad (3)$$

Finally, $\hat{x}_0$ is used as the final adversarial video clip $\hat{x}$ to fool the target video recognition model $F_\theta$.

**Preservation of Structural Similarity.** Adversarial optimization at each denoising step leads to a deviation of the latent from the original frame distribution. Despite the inevitable alterations to the benign frames for adding adversarial content, the challenge lies in preserving the structural similarity of the adversarial frames from the benign frames. Leveraging the fact that the spatial features of the self-attention layers are influential in determining both the structure and the appearance of the generated images, TALO minimizes the average difference of the self-attention maps between the benign and the adversarial latent at each timestep $t$:

$$\arg\min_{\hat{x}_t} \mathcal{L}_{structure} = \sum_{j \in n_s} ||\hat{s}_t^j - s_t^j||_2^2 \quad (4)$$

where $s_t^j$, $\hat{s}_t^j$ are respectively the $j$-th self-attention map of benign latents $x_t$ and adversarial latents $\hat{x}_t$, $n_s$ denotes the total number of self-attention maps in the diffusion model.

In general, the final objective function of ReToMe-VA is as follows, where $\gamma$ and $\beta$ represent the weight factors of each loss:

$$\arg\min_{\hat{x}_t} \mathcal{L}_{total} = \gamma\mathcal{L}_{attack} + \beta\mathcal{L}_{structure} \quad (5)$$

**Incremental Iteration Strategy.** TALO iteratively optimizes $\hat{x}_t$ to seek optimal adversarial latents at timestep $t$ and the iteration number represents a trade-off between spatial imperceptibility and adversarial transferability. Recent work [22] has indicated that the diffusion models tend to add coarse semantic information (e.g., layout) during the early timesteps while more fine details during the later timesteps. As depicted in Table 6, a smaller number of iterations fail to find better perturbations, reducing the low adversarial transferability. Conversely, a larger number of iterations render adversarial frames deviating more from the benign frames, adversely affecting the spatial imperceptibility of the adversarial video clip. Therefore, we adopt an Incremental Iteration (II) strategy, starting with fewer attack iterations during the early timesteps to preserve structure and gradually increasing the number of iterations during the later timesteps to add adversarial details.

Our TALO strategy has two advantages. First, timestep-wise optimization with II strategy provides a more controllable and stable process during adversarial generation making more powerful

---

**Algorithm 1:** Framework of ReToMe-VA

**Input:** a benign video clip $x$ with label $y$, a surrogate classifier $G_\phi$, DDIM steps $T$, start attack DDIM timestep $t_s$, initial attack iteration $N_a$, recursive token merging ratio $p$, weight factors $\gamma$, $\beta$.
**Output:** Unrestricted adversarial video clip $\hat{x}$.

1 Add Recursive Token Merging and Recursive Token Unmerging Module to Stable Diffusion;
2 Calculate latents $\{x_1, ..., x_{t_s}\}$ using DDIM inversion;
3 $\hat{x}_{t_s} \leftarrow x_{t_s}$;
4 **for** $t \leftarrow t_s$ **to** 1 **do**
5     **for** $j \leftarrow 1$ **to** $N_a + 2(t_s - t)$ **do**
6         $\hat{x}_0^t = \frac{\hat{x}_t - \sqrt{1 - \alpha_t}\epsilon_\theta(\hat{x}_t, t)}{\sqrt{\alpha_t}}$;
7         Calculate the attack loss $\mathcal{L}_{attack}$ as Eq. 2;
8         Calculate the structure loss $\mathcal{L}_{structure}$ as Eq. 4;
9         Update $\hat{x}_t$ over total loss $\mathcal{L}_{total}$ Eq. 5 with AdamW optimizer;
10         $\hat{x}_{t-1} \leftarrow$ Eq. 3 ;
11 $\hat{x} \leftarrow \hat{x}_0$;
12 **return** $\hat{x}$

---

adversarial video clips with spatial imperceptibility. Second, TALO only involves one timestep gradient computation thereby reducing memory consumption in gradient computation.

## 3.3 Recursive Token Merging

TALO strategy perturbs each benign frame of video separately. This per-frame optimization makes the frames likely optimized along different adversarial directions resulting in motion discontinuity and temporal inconsistency. Furthermore, separately perturbing each benign frame reduces the monotonous gradients because the interactions among the frames are not exploited. To this end, we introduce a recursive token merging (ReToMe) strategy that recursively matches and merges similar tokens across frames together enabling the self-attention module to extract consistent features. In the following, we first provide the basic operation of token merging and token unmerging and then our recursive token merging algorithm.

Generally, tokens $T$ are partitioned into a source (src) and destination (dst) set. Then, tokens in src are matched to their most similar token in dst, and $r$ most similar edges are selected subsequently. Next, we merge the connected $r$ most similar tokens in src to dst by replacing them as the linked dst tokens. To keep the token number unchanged, we divide merged tokens after self-attention by assigning their values to merged tokens in src. Token matching, merging, and unmerging operations are expressed as:

$$e = match(src, dst, r), \\ T_m = M(T, e), T_{um} = UM(T_m, e). \quad (6)$$

where $match(\cdot)$ outputs the matching map $e$ with $r$ edges from src to dst, $M(\cdot)$ and $UM(\cdot)$ merge and unmerge tokens according to matching $e$. After token merging operation, $T_m = \{(T^{src})^{um}, T^{dst}\}$ consists the unmerged tokens $(T^{src})^{um}$ in src and tokens $T^{dst}$ in

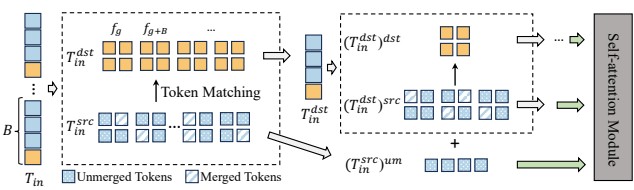

**Figure 3: Recursive token merging process.**

$dst$, while merged tokens $(T^{src})^m$ in $src$ is replaced by tokens in $dst$.

A self-attention module takes a sequence of input and output tokens across all frames. The input and output tokens are denoted as $T_{in}, T_{out} \subset \mathbb{R}^{N \times L \times E}$, where $L$ is the number of tokens per frame, and $E$ is the embedding dimension. To partition tokens across frames into $src$ and $dst$, we define stride as $B$, we randomly choose one out of the first $B$ frames (e.g. the $g^{th}$ frame), and select the subsequent frames every B interval into the $dst$ set (named as $T_{in}^{dst}$). Tokens of other frames are in $src$ set ($T_{in}^{src}$). Then merging operation mentioned above in Eq. 6 is used to merge source frames:

$$e_1 = match(T_{in}^{src}, T_{in}^{dst}, r_1),$$
$$T_{rm} = M(T_{in}, e1).$$
(7)

where $T_{rm} = \{T_{in}^{dst}, (T_{in}^{src})^{um}\}$. We set $r_1 = p(N - N_{d_1})L$ where $p$ is the merging ratio, $(N - N_{d_1})L$ is the $src$ token number in the first merging process and $N_{d_1}$ is the $T_{in}^{dst}$ frame number.

Nevertheless, during the merging process expressed above, tokens in $dst$ are not merged and compressed. To maximally fuse the inter-frame information, we recursively apply the above merging process to tokens in $dst$ until they contain only one frame. For instance, in the next merging process of $T_{in}^{dst}$, after partition of $src$ and $dst$ of $T_{in}^{dst}$ (named as $(T_{in}^{dst})^{src}$ and $(T_{in}^{dst})^{dst}$), we merge tokens in $src$ to $dst$ by:

$$e_2 = match((T_{in}^{dst})^{src}, (T_{in}^{dst})^{dst} + (T_{in}^{src})^{um}, r_2),$$
$$(T_{in}^{dst})_{rm} = M(T_{in}^{dst}, e_2).$$
(8)

We set $r_2 = p(N_{d_1} - N_{d_2})L$ where $(N_{d_1} - N_{d_2})L$ is the $src$ token number and $N_{d_2}$ is $dst$ frame number in this process. The difference is that we add the previous unmerged tokens $(T_{in}^{src})^{um}$ into $dst$ for token matching. Then we replace $T_{in}^{dst}$ with $(T_{in}^{dst})_{rm}$ in $T_{rm}$. The token merging process of ReToMe is shown in Figure 3. Next, we input the tokens $T_{rm}$ into the self-attention module to calculate $(T_{out})_{rm}$.

The output tokens $(T_{out})_{rm}$ need to be restored to their original shape $T_{out}$ to perform the following operations. Therefore, in the unmerge process, the unmerging operation in Eq. 6 is applied in the reverse order of the merging process to get $T_{out}$.

Our ReToMe has three advantages. Firstly, ReToMe ensures that the most similar tokens share identical outputs, maximizing the compression of tokens. This approach fosters internal uniformity of features across frames and preserves temporal consistency, thereby effectively achieving temporal imperceptibility. Secondly, given the fact that there is a negative correlation between the adversarial transferability and the interaction inside adversarial perturbations [35], the merged tokens decrease interaction inside adversarial

perturbations, effectively preventing overfitting on the surrogate model. Furthermore, the tokens in $dst$ linked to merged tokens facilitate inter-frame interaction in gradient calculation, which may induce more robust and diverse gradients [34]. Therefore, ReToMe can effectively boost adversarial transferability.

## 4 EXPERIMENT

### 4.1 Experiment Settings

**Dataset.** We evaluate the adversarial transferability of our proposed method on Kinetics-400 [4] dataset. The dataset contains approximately 240,000 videos from 400 human action classes, we carefully selected one video clip from each class that was correctly classified by all video recognition models, yielding a total of 400 videos as the validation dataset.

**Models.** To assess the adversarial robustness of network architectures, we select CNNs and ViTs as the attacked models, respectively. For CNNs, we choose normally trained I3D SLOW [9], TPN [42] with two different backbones: ResNet-50 and ResNet-101, and R(2+1)D [33] with backbone ResNet-50 (R(2+1)D-50). For ViTs, we consider VTN [24], Motionformer [2], TimeSformer [26], Video Swin [19].

**Implementation Details.** Our experiments are run on an NVIDIA A800 with Pytorch. We set DDIM steps $T = 20$, start attack DDIM step $t_s = 5$, initial attack Iteration $N_a = 4$, recursive token merging ratio $p = 0.5$. Meanwhile, the weight factors $\gamma, \beta$ in Eq. 5 are set to 10, 100 respectively. We adopt AdamW [20] with the learning rate set to $1e^{-2}$. The version of Stable Diffusion we used is v2.0.

**Evaluation Metrics.** We use the Attack Success Rate (ASR), i.e., the percentage of adversarial video clips that are successfully misclassified by the video recognition model, to evaluate the adversarial transferability. Thus a higher ASR means better adversarial transferability. If not specifically stated, Avg.ASR is the average ASR over all target video models. Besides, we quantitatively assess the frame quality using two reference perceptual image quality measures including Frechet Inception Distance (FID) [14] and LPIPS [44], and three non-reference perceptual image quality measures NIMA-AVA [32], HyperIQA [31], and TReS [10]. For temporal consistency, we adopt four evaluation metrics in VBench [16], including Subject Consistency, Background Consistency, Motion Smoothness, and Temporal Flickering. Each metric is tailored to specific aspects of video analysis. Subject Consistency measures whether an object's appearance remains consistent throughout the video. Background Consistency evaluates the temporal uniformity of background scenes through CLIP [27] feature similarity across frames. Motion Smoothness assesses the smoothness and realism of motion, adhering to real-world physics. Temporal Flickering computes the mean absolute difference across frames to detect abrupt changes. Moreover, we also select Pixel-MSE to evaluate the naturalness and continuity of frame-to-frame transitions. Specifically, each frame in the adversarial video clip is warped to the next frame by the optical flow between consecutive frames. Then, we compute the average mean-squared pixel error between each warped frame and its corresponding next frame.

**Table 1: Performance comparison of adversarial transferability on normally trained CNNs and ViTs. We report attack success rates (%) of each method ("*" is white-box attack results). The best results are highlighted in bold.**

| Surrogate Model | Attack | Models | | | | | | | | | Avg. ASR (%) |
| --- | --- | --- | --- | --- | --- | --- | --- | --- | --- | --- | --- |
| | | CNNs | | | | | Transformers | | | | |
| | | Slow-50 | Slow-101 | TPN-50 | TPN-101 | R(2+1)D-50 | VTN | Motionformer | TimeSformer | Video Swin | |
| Slow-50 | TT | 99.00* | 74.00 | **96.50** | 72.00 | 66.25 | 5.50 | 3.50 | 6.75 | 10.75 | 41.91 |
| | SAE | 37.75* | 9.00 | 12.75 | 8.50 | 60.50 | 14.00 | 22.25 | 37.75 | 21.25 | 20.41 |
| | ReColorAdv | 100.00* | 64.50 | 96.25 | 56.25 | 68.00 | 7.25 | 4.75 | 13.25 | 11.75 | 40.25 |
| | cAdv | 98.75* | 29.00 | 43.25 | 30.00 | 28.25 | 25.00 | 21.50 | **44.25** | 24.25 | 30.69 |
| | tAdv | 99.50* | 7.00 | 13.25 | 7.50 | 36.00 | 4.50 | 2.75 | 9.25 | 6.25 | 10.81 |
| | ACE | 89.25* | 3.75 | 6.50 | 4.25 | 24.00 | 3.25 | 4.00 | 9.75 | 4.75 | 7.53 |
| | ColorFool | 31.75* | 5.25 | 9.50 | 7.50 | 50.25 | 11.50 | 19.25 | 30.75 | 17.50 | 16.62 |
| | NCF | 37.25* | 12.25 | 21.25 | 10.50 | 54.00 | 12.00 | 15.50 | 25.00 | 13.25 | 18.38 |
| | ACA | 67.75* | 38.50 | 47.75 | 36.00 | **68.75** | 25.00 | 22.50 | 32.75 | 28.25 | 37.44 |
| | Ours | 96.50* | **78.50** | 89.50 | **77.00** | 61.25 | **30.25** | **25.25** | 39.50 | **35.50** | 54.59 |
| TPN-50 | TT | 92.00 | 52.50 | 100.00* | 53.25 | 63.50 | 4.75 | 2.25 | 8.25 | 8.25 | 35.59 |
| | SAE | 9.00 | 7.00 | 36.25* | 6.50 | 59.00 | 14.50 | 21.50 | 40.25 | 21.50 | 19.56 |
| | ReColorAdv | 67.00 | 27.25 | 100.00* | 27.75 | 56.75 | 3.50 | 2.25 | 8.25 | 5.50 | 24.78 |
| | cAdv | 31.50 | 18.75 | 98.25* | 21.50 | 28.75 | 22.00 | 17.75 | 39.50 | 19.25 | 24.88 |
| | tAdv | 12.25 | 7.00 | 98.00* | 6.50 | 33.50 | 6.25 | 3.00 | 9.00 | 6.25 | 10.47 |
| | ACE | 4.00 | 3.50 | 86.75* | 2.75 | 22.00 | 4.25 | 3.75 | 10.50 | 4.75 | 6.94 |
| | ColorFool | 8.75 | 6.00 | 35.00* | 5.75 | 45.50 | 8.75 | 17.50 | 28.50 | 14.50 | 15.59 |
| | NCF | 20.25 | 10.25 | 32.00* | 9.75 | 53.75 | 10.75 | 14.75 | 26.50 | 12.25 | 17.06 |
| | ACA | 43.75 | 33.25 | 63.75* | 33.50 | **67.00** | 24.00 | 22.75 | 32.75 | 27.50 | 35.56 |
| | Ours | **80.50** | **58.75** | 97.50* | **61.75** | 52.75 | 20.75 | 19.75 | **33.00** | 27.25 | 44.31 |
| VTN | TT | 11.25 | 10.00 | 10.50 | 5.50 | 56.00 | 100.00* | 64.50 | 83.50 | 14.25 | 31.94 |
| | SAE | 8.75 | 6.25 | 9.00 | 7.25 | 55.00 | 48.75* | 19.00 | 39.75 | 22.50 | 20.16 |
| | ReColorAdv | 4.50 | 4.50 | 5.75 | 4.25 | 42.50 | 100.00* | 43.75 | 62.00 | 10.50 | 22.22 |
| | cAdv | 16.25 | 14.50 | 16.50 | 17.25 | 28.00 | 99.75* | 38.50 | 67.25 | 27.00 | 16.28 |
| | tAdv | 7.25 | 6.00 | 7.75 | 5.25 | 32.25 | 94.00* | 14.75 | 28.50 | 9.75 | 13.94 |
| | ACE | 3.00 | 2.00 | 3.00 | 2.00 | 22.75 | 71.25* | 5.50 | 18.50 | 3.50 | 7.53 |
| | ColorFool | 5.75 | 5.25 | 9.00 | 5.50 | 40.00 | 41.50* | 18.50 | 30.75 | 15.50 | 19.08 |
| | NCF | 16.50 | 10.75 | 15.75 | 9.75 | 53.75 | 72.25* | 24.75 | 39.25 | 14.00 | 21.66 |
| | ACA | 28.75 | 28.00 | 28.75 | 25.50 | **66.75** | 59.50* | 32.00 | 42.00 | 28.75 | 35.06 |
| | Ours | 27.25 | 25.25 | 28.25 | 23.00 | 49.00 | 99.25* | **75.50** | **88.25** | **43.25** | 44.97 |
| Motionformer | TT | 12.75 | 12.50 | 11.00 | 8.00 | 57.75 | 91.75 | 100.00* | 86.50 | 29.50 | 38.72 |
| | SAE | 7.75 | 4.50 | 6.75 | 4.25 | 49.50 | 11.50 | 72.00* | 31.75 | 14.00 | 16.25 |
| | ReColorAdv | 2.50 | 1.50 | 3.25 | 2.00 | 36.00 | 15.50 | 100.00* | 25.50 | 2.00 | 11.03 |
| | cAdv | 9.00 | 7.25 | 9.00 | 9.00 | 21.00 | 25.00 | 89.25* | 48.50 | 12.25 | 17.62 |
| | tAdv | 12.75 | 12.00 | 13.00 | 12.00 | 38.00 | 12.25 | 51.50* | 20.75 | 11.50 | 16.53 |
| | ACE | 1.75 | 1.75 | 2.25 | 0.25 | 6.00 | 0.75 | 50.00* | 6.50 | 2.25 | 2.69 |
| | ColorFool | 3.50 | 2.75 | 5.50 | 4.50 | 33.00 | 5.00 | 71.50* | 26.00 | 8.00 | 11.03 |
| | NCF | 12.50 | 9.25 | 15.00 | 7.50 | 53.25 | 12.75 | *39.75 | 30.25 | 12.50 | 17.44 |
| | ACA | 27.00 | 27.50 | 25.75 | 24.50 | **65.75** | 31.50 | 67.75* | 37.75 | 24.50 | 33.03 |
| | Ours | **42.50** | **44.25** | **44.25** | **42.75** | 57.50 | 91.25 | 100.00* | **91.00** | **63.75** | 59.66 |
| TimeSformer | TT | 10.75 | 10.00 | 10.25 | 6.25 | 57.00 | **85.25** | 57.25 | 100.00* | 16.00 | 31.59 |
| | SAE | 5.00 | 3.75 | 4.75 | 3.50 | 43.75 | 8.00 | 14.75 | 72.50* | 14.75 | 12.28 |
| | ReColorAdv | 7.50 | 6.75 | 7.00 | 5.25 | 49.25 | 59.00 | 38.50 | 100.00* | 10.00 | 22.91 |
| | cAdv | 10.50 | 11.25 | 12.00 | 10.00 | 23.25 | 43.25 | 31.00 | 100.00* | 24.25 | 20.69 |
| | tAdv | 5.50 | 5.00 | 5.50 | 4.50 | 30.50 | 17.00 | 10.25 | 95.00* | 7.00 | 10.66 |
| | ACE | 3.00 | 2.75 | 3.75 | 1.00 | 18.00 | 4.50 | 3.25 | 89.75* | 3.50 | 4.97 |
| | ColorFool | 5.25 | 3.00 | 5.00 | 2.75 | 33.25 | 5.00 | 8.50 | 65.75* | 8.50 | 8.91 |
| | NCF | 16.50 | 10.00 | 17.00 | 9.75 | 53.00 | 21.50 | 27.75 | 92.75* | 17.75 | 29.56 |
| | ACA | 30.75 | 28.25 | 29.50 | 27.00 | **67.00** | 46.00 | 36.00 | 72.25* | 30.25 | 36.84 |
| | Ours | 28.00 | **29.50** | **32.00** | **28.50** | 49.75 | 85.00 | **76.50** | 100.00* | **47.00** | 47.03 |

## 4.2 Attacks against Normally Trained Models

We first assess the adversarial transferability of normally trained CNNs and ViTs. For video restricted attacks, we compare the proposed method with state-of-the-art TT [37]. For video unrestricted attacks, due to the lack of comparable work, we extend the image unrestricted attacks to generate adversarial video clips frame-by-frame, including SAE [15], ReColorAdv [17], cAdv [3], tAdv [3], ACE [45], ColorFool [29], NCF [43], and ACA [7]. Adversarial video clips are crafted against Slow-50, TPN-50, VTN, Motionformer and TimeSformer respectively. The transferability of different methods is displayed in Table 1.

It can be observed that adversarial video clips generated by ReToMe-VA generally exhibit superior transferability compared to those generated by state-of-the-art competitors. Our proposed ReToMe-VA achieved a white-box attack success rate of 100% on the Motionformer and TimeSformer models. The results from Table 1

indicate that our method surpasses the restricted attack method TT in the black-box setting. When Slow-50, Motionformer, and TimeSformer are used as surrogate models, we significantly outperform state-of-the-art ACA by 17.10%, 26.62%, and 10.19%, respectively, indicating that our ReToMe-VA has higher transferability under the more challenging cross-architecture setting. Specifically, when the surrogate model is Slow-50, we surpass ACA by significant margins of 40%, 41.75%,41%, and 6.75% in Slow-101, TPN-50, TPN-101, and TimeSformer, respectively.

## 4.3 Attacks against Adversarial Defense Mechanisms

We also assess its performance against five representative defense mechanisms, including the top-2 defense methods in the NIPS 2017 competition (high-level representation guided denoiser (HGD) [18] and random resizing and padding (R&P) [40]), three popular input

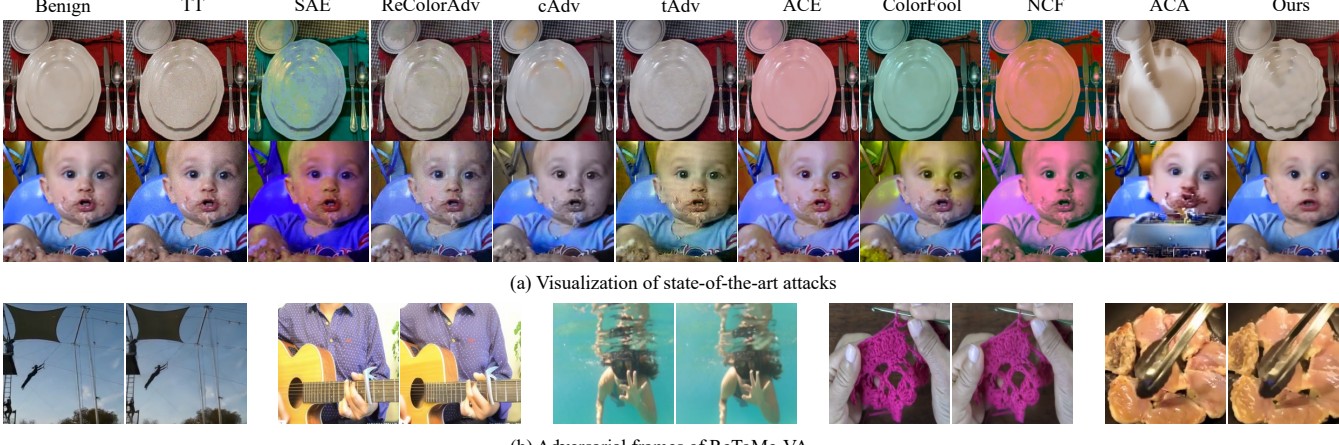

(a) Visualization of state-of-the-art attacks

(b) Adversarial frames of ReToMe-VA

**Figure 4: Qualitative results of frame quality. (a) Visual quality comparisons among different attack methods. (b) More adversarial frames generated from ReToMe-VA. The Left is the benign frame and the right is the adversarial frame.**

**Table 2: Robustness on adversarial defense methods. We report Avg.ASR(%) of each method. The best results are in bold.**

| Attack Method | HGD | R&P | JPEG | Bit-Red | DiffPure |
|---|---|---|---|---|---|
| TT | 37.69 | 29.47 | 31.69 | 38.56 | 12.59 |
| SAE | 24.03 | 25.00 | 26.34 | 27.81 | 37.31 |
| ReColorAdv | 35.81 | 29.13 | 29.84 | 35.53 | 15.69 |
| cAdv | 31.31 | 30.19 | 32.00 | 34.03 | 38.09 |
| tAdv | 10.00 | 10.63 | 11.28 | 15.34 | 15.72 |
| ACE | 8.09 | 9.31 | 10.40 | 12.84 | 20.71 |
| ColorFool | 18.88 | 20.50 | 21.25 | 22.94 | 33.56 |
| NCF | 20.69 | 22.25 | 21.69 | 24.75 | 32.16 |
| ACA | 35.90 | 28.22 | 29.84 | 35.53 | 36.56 |
| Ours | **53.41** | **50.97** | **52.72** | **54.56** | **40.97** |

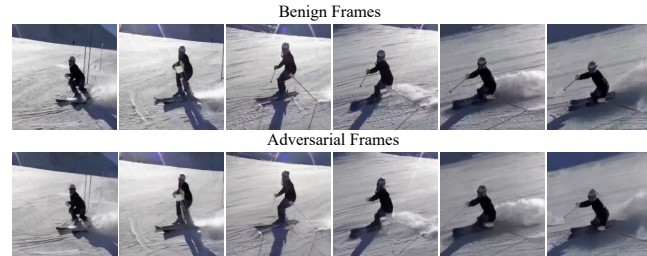

**Figure 5: A Sample of generated video from our method.**

pre-process defenses, namely jpeg compression (JPEG) [12], bit depth reduction (Bit-Red) [41], and DiffPure [25]. We take Slow-50 as a surrogate model and all of the adversarial video clips are crafted on it.

From the results demonstrated in Table 2, we can see our method displays superiority over other advanced attacks by a significant margin. For example, against HGD and DiffPure defenses, our method outperforms the next best attack ACA by over 17.5% and 4.41% respectively, indicating its robustness and efficiency in penetrating these defenses. This evidences the advanced capability of our method in maintaining high attack success rates under diverse adversarial defense methods.

## 4.4 Visualization

In this section, we will demonstrate the superiority of our approach through qualitative and quantitative comparisons of frame quality and temporal consistency in videos.

**Frame Quality.** In Figure 4(a), we visualize the adversarial frames crafted by different attack approaches. We can see that our attack is much more natural than the restricted attack TT and more imperceptible compared with other unrestricted attacks. In detail, the color and texture changes of adversarial frames generated by SAE, ACE, ColorFool, NCF, and ACA are easily perceptible. Next, we give more adversarial frames generated by ReToMe-VA in Figure 4(b). It is observed that our method adaptively modifies inconspicuous details to generate adversarial frames. For example, minor alteration is made to the texture of the knitted yarn in the frame in the fourth column of Figure 4(b). Moreover, we quantitatively assess the frame quality using the reference and non-reference perceptual image quality measures. As illustrated in Table 4, our method achieves top-2 performance across all metrics. And ReToMe-VA achieves the best result in HyperIQA and TReS.

**Temporal Consistency.** To provide a qualitative comparison, Figure 5 shows an adversarial video clip crafted by our ReToMe-VA. From the visualization of the video, we can observe that our proposed method produces high-quality frames. The crafted frames by ReToMe-Va highly align with the benign frames in both appearance and structure and also maintain a high level of motion consistency with the benign frames. Quantitative evaluation results are shown in Table 3, we evaluate the temporal quality of the videos using five metrics, all of which achieve top-2 results. Specifically, Motion

**Table 3: Quantitative comparison of temporal consistency. The best results are in bold and the second-best results are underlined.**

| Attack Method | Subject Consistency↑ | Background Consistency↑ | Motion Smoothness↑ | Temporal Flickering↑ | Pixel-MSE↓ |
|---|---|---|---|---|---|
| SAE | 79.23% | 87.08% | 82.61% | 80.61% | 94.17 |
| ReColorAdv | 87.69% | 91.72% | 95.07% | 93.00% | 69.99 |
| cAdv | 86.43% | 90.62% | 94.28% | 92.31% | 67.56 |
| tAdv | **88.81%** | **93.29%** | 95.50% | 93.44% | **57.50** |
| ACE | 85.03% | 91.83% | 92.27% | 90.19% | 85.01 |
| ColorFool | 78.94% | 88.29% | 79.44% | 76.88% | 83.81 |
| NCF | 79.82% | 89.37% | 87.65% | 85.02% | 95.58 |
| ACA | 75.67% | 85.89% | 94.10% | 91.96% | 68.98 |
| Ours | 88.03% | 92.21% | **95.62%** | **93.76%** | 58.66 |

Smoothness and Temporal Flickering yield the best results. Therefore, our method demonstrates superior performance in terms of video temporal consistency.

**Table 4: Quantitative evaluation of image quality. The best results are highlighted in bold while the second-best results are underlined. NA denotes Not Applicable.**

| Attack Method | FID↓ | LPIPS↓ | NIMA-AVA↑ | HyperIQA↑ | TReS↑ |
|---|---|---|---|---|---|
| Benign | NA | NA | 5.38 | 50.97 | 59.80 |
| TT | 43.15 | 0.13 | 5.46 | 50.81 | 58.08 |
| SAE | 57.66 | 0.39 | **5.64** | 49.61 | 57.22 |
| ReColorAdv | 50.40 | 0.13 | 5.46 | 50.81 | 58.08 |
| cAdv | 47.02 | 0.20 | 5.61 | 52.58 | 61.41 |
| tAdv | 36.75 | **0.08** | 5.37 | 49.46 | 57.30 |
| ACE | **21.63** | 0.13 | 5.31 | 51.28 | 59.92 |
| ColorFool | 48.79 | 0.38 | 5.18 | 50.13 | 58.98 |
| NCF | 37.02 | 0.32 | 5.18 | 48.95 | 54.95 |
| ACA | 41.69 | 0.24 | 5.60 | 48.74 | 55.86 |
| Ours | 25.63 | 0.10 | 5.62 | **55.53** | **66.31** |

## 4.5 Ablation Studies

In Table 5, we ablate the designs mentioned in Section 3.3. We can observe that the avg.ASR and Subject Consistency increase by 6.08% and 0.04 by using ReToMe, indicating that the Recursive Token Merging Technique exhibits strong adversarial transferability and enhanced temporal consistency. Additionally, the ablation study of II strategy is shown in Table 6. In detail, the first two lines denote that we fix the iteration number at each timestep, while the last line displays our II strategy. The results verify that our II strategy performs a good trade-off between transferability and spatial imperceptibility.

| ReToMe | Avg.ASR | Subject Consistency |
|---|---|---|
| w/o | 53.17 | 0.8410 |
| w/ | 59.25 | 0.8803 |

| Iter Strategy | Avg. ASR (%) | FID |
|---|---|---|
| Fix Iter 4 | 44.69 | 18.86 |
| Fix Iter 12 | 70.11 | 33.42 |
| Iter 4→12 | 59.25 | 25.63 |

**Table 5: Ablation study of Re-ToMe.**
**Table 6: Ablation study of II strategy.**

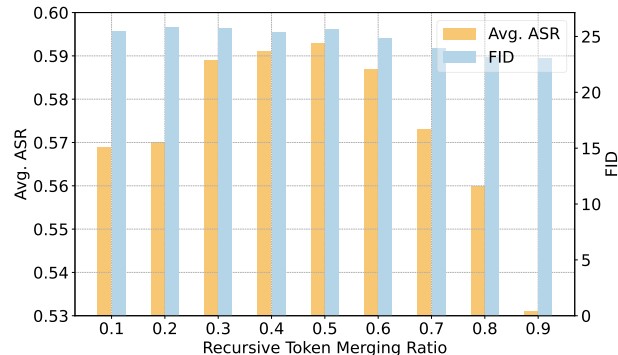

**Figure 6: Comparison of different merging ratios.**

Moreover, we investigate the impact of different merging ratios on adversarial transferability and video quality, using Slow-50 as an example surrogate model. The results are illustrated in Figure 6, which demonstrate that a merging ratio of $p = 0.5$ achieves the best adversarial transferability with high frame quality.

## 5 CONCLUSION

In this paper, we propose the Recursive Token Merging for Video Diffusion-based Unrestricted Adversarial Attack (ReToMe-VA). As far as we know, this is the first diffusion-based framework to generate imperceptible adversarial video clips with higher transferability. ReToMe-VA adopts a Timestep-wise Adversarial Latent Optimization strategy to achieve spatial imperceptibility. Moreover, ReToMe-VA introduces a Recursive Token Merging (ReToMe) mechanism. By aligning and compressing redundant tokens across frames, ReToMe produces temporally consistent adversarial videos. ReToMe provides more diverse and robust attack direction by incorporating inter-frame interactions into the adversarial optimization process, consequently boosting adversarial transferability. Extensive experiments and visualization demonstrate the efficacy of ReToMe-VA, particularly in surpassing the best baseline by an average of 14.16% in normally trained models. We hope our work will pave the way for future research in enhancing the robustness of video recognition models against adversarial threats, as well as contributing to the development of more effective video adversarial attack methods.

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
