# OpenReview forum: "ReToMe-VA: Recursive Token Merging for Video Diffusion-based Unrestricted Adversarial Attack"
_acmmm.org/ACMMM/2024/Conference — MM2024 Poster_

### Official Review · Reviewer_XYwS · 2024-05-22

**Rating:** 4
**Confidence:** 3

**Summary:**

Authors proposed a Recursive Token Merging for Video Diffusion-based Unrestricted Adversarial Attack, namely ReToMe-VA. Proposed framework can generate imperceptible adversarial video clips with enhanced transferability. The article is well written, but some of the technical details are not presented clearly. Please see the list below for details.

**Strengths:**

+ The paper is well organized
+ The issues of transferability and continuity that the paper targeted are meaningful for the study of video adversarial examples
+ Evaluation is comprehensive

**Limitations:**

- line 342 where x_0 is x, what is the x? is this a typo? Should it be T here?
- line 344 is the t_s selected by attacker? or the last b frame?
- The incremental iteration strategy is very conceptual. How do you adjust the number of configuration iterations for actual use? This was also not discussed in the evaluation.
- line 394 there is no table 6 in paper and supply
- sec.3.3 The recursive token merge technique is not clearly explained:
-- How are tokens defined?
-- How is similarity measured?
-- The explanation in the main text does not match Figure 3. It is recommended that the authors provide additional explanations for this section and add references if applicable.
- Recursive Token Merging makes a trade-off for better transferability as well as frame content coherence. This meets the expectations of Table 5 for consistency, but also better ASR, why?
- How does the recursive token merge ratio affect performance? Why is half the merge ratio best? It's worth a deeper discussion here.

**Suitability:**

3

---

### Official Review · Reviewer_qJYP · 2024-05-23

**Rating:** 4
**Confidence:** 4

**Summary:**

The paper proposes a framework called ReToMe-VA for generating imperceptible adversarial video clips with higher transferability.  The framework incorporates the Timestep-wise Adversarial Latent Optimization (TALO) strategy to achieve spatial imperceptibility and the Recursive Token Merging (ReToMe) mechanism to achieve temporal imperceptibility.

**Strengths:**

(1) The paper addresses the gap in the literature by proposing the first framework for video diffusion-based unrestricted adversarial attacks.
(2) The TALO strategy and ReToMe mechanism are innovative and effective in achieving spatial and temporal imperceptibility in adversarial videos.

**Limitations:**

(1) The evaluation of the proposed method is limited.  The paper only provides results on the Kinetics-400 dataset (why not other video datasets such as UCF101, ActivityNet) and does not compare the proposed method with other state-of-the-art approaches in the field.  This lack of comparison makes it difficult to assess the novelty and effectiveness of the proposed method.
(2) The paper does not provide a thorough analysis of the experimental results.  There is no discussion of the limitations or potential drawbacks of the proposed method, and there is no analysis of the failure cases or potential reasons for the low success rates.

**Suitability:**

3

---

### Official Review · Reviewer_kp78 · 2024-05-24

**Rating:** 5
**Confidence:** 3

**Summary:**

This paper presents a novel framework called Recursive Token Merging Unrestricted Adversarial Attack (ReToMe-VA) based on video diffusion. Its main focus is to generate antagonistic video clips that are imperceptible and highly transferable. Unlike existing research that mainly focuses on images, the framework specifically targets adversarial attacks in the video context. ReToMe-VA utilizes Timestep-wise Adversarial Latent Optimization (TALO) and Recursive Token Merging (ReToMe) to achieve imperceptibility and transferability. Extensive experiments demonstrated the advantages of ReToMe over the state-of-the-art methods.

**Strengths:**

1. The idea of applying diffusion model to generate unrestricted video adversarial examples is novel. Previous research has extensively explored unrestricted image adversarial attacks, but fewer studies have been conducted in the video domain. This paper identifies two challenges in generating unrestricted video adversarial attacks, namely temporal consistency and model transferability, and provides two solutions to address these challenges.

2. This paper is technically sound since diffusion models have been used in the image domain to generate adversarial examples. Compared to the distance-regularized form, unrestricted adversarial examples show better imperceptibility and transferability.

3. Two strategies are introduced in this paper to further improve the stealth and transferability of the attack, which is supported by extensive experimental results. The transferability of adversarial examples generated by the proposed method is better than 9 baseline attacks in most cases. Meanwhile, visualized examples demonstrate the high quality of generated examples.

**Limitations:**

A major concern is about the selection of video adversarial defense methods, the paper evaluated the proposed method under 5 defenses. However, 4/5 of these methods are proposed before 2020 and all of them were initially designed for image adversarial instances. Therefore, it would be useful for the authors to explain why. Also, several video adversarial defense methods have been proposed in the past few years (e.g., [1-3]), so it would be desirable to use these new defense methods for discussion and analysis.

[1] Lo, Shao-Yuan, and Vishal M. Patel. "Defending against multiple and unforeseen adversarial videos." IEEE Transactions on Image Processing 31 (2021): 962-973.

[2] Kinfu, Kaleab A., and René Vidal. "Analysis and extensions of adversarial training for video classification." Proceedings of the IEEE/CVF Conference on Computer Vision and Pattern Recognition. 2022.

[3] Lee, Hong Joo, and Yong Man Ro. "Defending Video Recognition Model against Adversarial Perturbations via Defense Patterns." IEEE Transactions on Dependable and Secure Computing (2023).

**Suitability:**

2

---

### Official Review · Reviewer_FCZs · 2024-05-24

**Rating:** 4
**Confidence:** 2

**Summary:**

This paper introduces an unrestricted adversarial attack method against Stable Diffusion to generate imperceptible adversarial videos with high transferability. The proposed method is based on a time-step-wise latent optimization strategy that gradually updates perturbations in latent space at each step. Additionally, a token merging mechanism is proposed to enhance adversarial transferability. A series of experiments demonstrate the effectiveness of the proposed method in various settings.

**Strengths:**

1. The paper is well-written and organized.
2. The proposed method is novel and effective.
3. Extensive experiments validate the effectiveness of the proposed method in diverse settings.
4. The visualization results are impressive.

**Limitations:**

1. It would be interesting to see if the proposed method can be extended to standard adversarial settings, such as the lp norm setting, possibly by applying a simple clip at the pixel level.
2. The authors should provide an error analysis, such as a confusion matrix, to give a more detailed understanding of the model's performance under attacks / natural samples.
3. The authors could also apply the proposed attack methods to adversarially trained models to evaluate their performance, as adversarial training is one of the most effective adversarial defense methods.
4. I noticed that the performance of the proposed method is not as good as tAdv. I would like the authors to provide further discussions about it. If possible, the authors could provide more comparisons in diverse settings.
5. The ablation study should also considers diverse combinations of losses.

**Suitability:**

3

---

### Meta-Review · Area_Chair_s53D · 2024-06-29

**Recommendation:** Accept (Poster)
**Confidence:** 5

**Metareview:**

This paper proposed the Recursive Token Merging for Video Diffusion-based Unrestricted Adversarial Attack which can generate imperceptible adversarial video clips with higher transferability. The paper received 4 acceptances from reviewers. AC has looked at reviews and agreed the acceptance of this paper.